Estimating uncertainty in density surface models

Miller David L. dave@ninepointeightone.net 1
Becker Elizabeth A. 2
Forney Karin A. 3 4
Roberts Jason J. 5
Cañadas Ana 5
Schick Robert S. 5
1 Centre for Research into Ecological & Environmental Modelling and School of Mathematics & Statistics, University of St Andrews , St Andrews , Fife , Scotland
2 Ocean Associates, Inc. under contract to Marine Mammal and Turtle Division, Southwest Fisheries Science Center National Marine Fisheries Service, National Oceanic and Atmospheric Administration , La Jolla , CA , United States of America
3 Marine Mammal and Turtle Division, Southwest Fisheries Science Center, National Marine Fisheries Service, National Oceanic and Atmospheric Administration , Moss Landing , CA , United States of America
4 Moss Landing Marine Laboratories, San Jose State University , Moss Landing , CA , United States of America
5 Marine Geospatial Ecology Laboratory, Nicholas School of the Environment, Duke University , Durham , NC , United States of America
Ward Eric
Electronic publication date: 2022 Aug 23
Publication date: 2022
Volume: 10
Electronic Location ID: e13950
Received 2022 Jun 30; Accepted 2022 Aug 5
Copyright year: 2022
Copyright holder: Miller et al.
License: This is an open access article, free of all copyright, made available under the Creative Commons Public Domain Dedication. This work may be freely reproduced, distributed, transmitted, modified, built upon, or otherwise used by anyone for any lawful purpose.
License URL: https://creativecommons.org/publicdomain/zero/1.0/

Keywords: Density surface models, Distance sampling, Uncertainty quantification, Spatial modelling, Species distribution modelling, Model uncertainty, Environmental uncertainty

Funding: OPNAV N45 and the SURTASS LFA Settlement Agreement, and being managed by the U.S. Navy’s Living Marine Resources program N39430-17-C-1982 This work was funded by OPNAV N45 and the SURTASS LFA Settlement Agreement, and being managed by the U.S. Navy’s Living Marine Resources program under Contract No. N39430-17-C-1982. The funders had no role in study design, data collection and analysis, decision to publish, or preparation of the manuscript.

==============================
Providing uncertainty estimates for predictions derived from species distribution models is essential for management but there is little guidance on potential sources of uncertainty in predictions and how best to combine these. Here we show where uncertainty can arise in density surface models (a multi-stage spatial modelling approach for distance sampling data), focussing on cetacean density modelling. We propose an extensible, modular, hybrid analytical-simulation approach to encapsulate these sources. We provide example analyses of fin whales Balaenoptera physalus in the California Current Ecosystem.

Introduction

Reliable estimates of uncertainty in abundance are essential for management and conservation of biological populations. One of the most popular methods of estimating abundance is distance sampling (Buckland et al., 2001), which uses data collected on the distances between sampler and observation to estimate the detection probability. This detection probability can be used in design-based estimates (via a Horvitz-Thompson-like estimator) or in model-based estimates to obtain abundance and density estimates. Here we focus on density surface models (DSMs; Hedley & Buckland, 2004; Miller et al., 2013), a model-based approach to obtain spatially-explicit abundance and density estimates. Spatially-explicit models allow managers and scientists to ask more fine-grained questions of their data. For example, marine species applications include spatial abundance estimation (Becker et al., 2016; Becker et al., 2017; Forney et al., 2015), habitat preference (Cañadas & Hammond, 2008; Torres, Read & Halpin, 2008; Hazen et al., 2017), spatial prioritization (Winiarski et al., 2014), risk assessment (Gilles et al., 2016; Redfern et al., 2013) and as a tool to assess potential impacts on cetaceans as required by government regulations (Roberts et al., 2016; Mannocci et al., 2017b). Ensuring that the major sources of uncertainty are included in final estimates allows managers and policy makers to make the best possible decisions by allowing them to evaluate the degree to which they can believe their results or the impact of proposed management actions. Failing to correctly account for uncertainty may lead to management strategies that have dire consequences, especially for endangered species.

Our interest in uncertainty quantification comes from the legal requirements surrounding the US Marine Mammal Protection Act of 1972 (MMPA), which bans the intentional “take” (disturbance or harm) of marine mammals but permits activities that may incidentally take them, provided that the number of takes is estimated with suitable methods and found to be sufficiently small. Uncertainty is built-in to the calculation of this limit (potential biological removal, PBR; e.g., Taylor et al., 2000) as it uses the 20% quantile of the distribution of abundance to give a minimum abundance estimate.

Every seven years the US Navy must apply for a “Letter of Authorization” to conduct peacetime testing and training activities that may take cetaceans, e.g., during the use of tactical SONAR. The US Navy uses simulations to assess the impact of sound from SONAR and other sources on cetaceans, and quantifies the take of each affected stock (US Department of the Navy, 2017). A primary input to these simulations are spatially explicit models of cetacean density (Roberts et al., 2016; Becker et al., 2016). In this and similar processes, it is critical that the major sources of uncertainty are clearly accounted for and propagated through the different phases of impact modelling.

In this article we propose an approach to uncertainty characterization and estimation that will help practitioners in two ways: (i) we give a checklist of the major sources of uncertainty, so possible pitfalls can be considered before a survey is conducted and appropriate field methods can be adapted in advance; (ii) we provide a framework to estimate the combined uncertainty from various sources, once data are collected.

In this article we adopt a modular, simulation-based approach to uncertainty estimation for DSMs. Rather than deriving a single, complex analytical expression, we use posterior simulation (sometimes referred to as “parametric bootstrapping”) where possible. Our approach uses the (posterior) distribution of the model parameters and samples from that distribution, from which we can obtain corresponding predictions. The variation in these predictions represents the uncertainty in our model. Simulation-based approaches can be easier to understand, less technically and notationally demanding, and more easily parallelizable. Our approach includes uncertainty from each model component while estimating covariance between components where possible. It is conceptually and computationally tractable, while avoiding potential pitfalls that occur with other approaches like the non-parametric bootstrap (see ‘Discussion’).

The article is structured as follows: ‘Density Surface Models’ gives a brief overview of the density surface modelling framework. ‘Characterising Components of Uncertainty’ then lists the various sources of uncertainty that may be present in these models and gives a summary of current methods to estimate uncertainty for each. ‘Combining Uncertainty from Multiple Model Components’ presents our framework for integrating uncertainty and we apply this to line transect survey data of fin whales (Balaenoptera physalus) in the California Current Ecosystem in ‘Example: Fin Whales in the California Current’. Extensions and future work are outlined in ‘Discussion’.

Density surface models

DSMs are multi-stage species distribution models (SDMs), where biases in the observation process(es) are first addressed, and then a spatial model is fitted to the resulting corrected data. An overview of DSM methodology is given in Miller et al. (2013) (see also Hedley & Buckland, 2004). A typical DSM might take into account changes in detectability resulting from the distance between sampler and observation using distance sampling methods (Buckland et al., 2001). A detection function is fitted to the distances between the sampler and collected observations, and from that detection function a probability of detection (unconditional on distance), p ˆ, is estimated. This probability is then used as an offset (along with the effort expended) in a generalized additive model (Wood, 2017), which might have the following mathematical form: (1) Enj=Ajp ˆj expβ0+fxyxj,yj+fSSTSSTj,

where j indexes sample units. In the case of point transects, j indexes the points; for line transects, the longer transects are cut into segments and then indexed by j (without loss of generality we refer to sample units as segments henceforth). Aj is the area of segment j, pj ˆ is the estimated probability of detection in that segment. nj is the number of observations (individuals or groups) in segment j. We assume that nj is distributed according to some count distribution (e.g., Tweedie or negative binomial) and β0 is the intercept. In this example fxy is a smooth function of location and fSST is a smooth of sea surface temperature; more generally, any number of smooth terms can be added (denoted f with subscript for the covariate).

We can adapt (Eq. (1)) to include information about other observation processes. These could include additional data collected either during the survey or during some other period. For example, we may want to include whether an animal is available to be detected (a problem for animals which dive under the water such as seabirds, cetaceans or pinnipeds). Availability can be addressed as another offset multiplier in Eq. (1), u ˆj. We may also want to relax the distance sampling assumption that animals on the trackline are detected with certainty, e.g., by applying mark-recapture distance sampling (Burt et al., 2014), in which we use data from multiple observers to estimate g(0), the probability of detecting an animal on the trackline in segment j. Again this quantity can be included in the offset of Eq. (1). We can extend Eq. (1) to include these additional estimates as follows: (2) Enj=Ajp ˆjg0 ^ju ˆj expβ0+fxyxj,yj+fSSTSSTj.

DSMs are commonly developed using multiple years of survey data, both to increase sample size and capture a wider range of environmental conditions, and then model predictions are made on finer temporal scales to evaluate seasonal and inter-annual differences in abundance and distribution (Becker et al., 2012; Forney et al., 2012; Redfern et al., 2019). For this reason, accounting for these biases is important, especially in the case where we are potentially combining data from multiple sources (e.g., Miller et al., 2021), as we need to ensure that any spatial heterogeneity that the GAM models is due to spatial effects, not down to unaddressed biases in the data. Model predictions can be made on days, weeks, or months, depending on the scale of the ecological question and the variability of the study ecosystem (Mannocci et al., 2017a).

Characterising components of uncertainty

Our aim is to think about uncertainty estimation in a modular way. We first give a taxonomy and our general strategies for including uncertainty from model components into our estimates, before moving on to review available methods for uncertainty estimation for the components we have seen so far.

We are generally concerned with two types of components: ones which have covariance with the spatial model and those which do not; we refer to these as coincident and non-coincident, respectively. Non-coincident components can use the delta method (Seber, 1987) and add their squared coefficient of variation to that of the model. This is easily justified if the estimate is from a different place and time (e.g., an estimate of availability from the literature for this species). The non-coincident case trivially includes the case where there are no covariates in that model component. If estimates are coincident in space and time, our general strategy here is to absorb this into the spatial model if possible, using the method of Bravington, Miller & Hedley (2021). A typical example is that detection functions will usually be coincident as they are estimated from data collected during the survey, in that case it is likely that weather conditions that affect detectability (e.g., sea state) vary in space, so there will be covariance between the detectability and spatial model that must be accounted for.

Methods for uncertainty estimation are more developed for some model components than others. We begin by looking at those which are fairly mature in some depth before moving onto areas that are still in development or have not been applied in the DSM context yet.

GAM (smooths, smoothing parameters, model structure)

Smooths in the spatial model (such as fxy and fSST) are estimated and so have associated model-based uncertainties from standard GAM theory, based on their basis function coefficients (β ˆ). We can also incorporate uncertainty in the smoothing parameter(s), λ, which dictate how wiggly the smooths should be.

GAMs fitted using the popular R package mgcv using restricted maximum likelihood (REML) are empirical Bayes estimates (Wood, 2017, Section 6.2.6), so we have an approximate posterior distribution β|λ∼Nβ ˆ,Vβ ˆ (where β ˆ are the estimated GAM coefficients and Vβ ˆ is their corresponding covariance matrix). Taking samples from this distribution we can make predictions of abundance, then make appropriate summaries over these predictions to obtain estimates of the variance, intervals, or other desired uncertainty statistics for the spatiotemporal extents that are useful for species management.

In order to move between the statements about parameters to statements about predictions (and then total abundance estimates) we form a matrix, X ~, that maps the model parameters to the linear predictor, η. We can then multiply X ~ by a sample from Nβ ˆ,Vβ ˆ to obtain predictions on the linear predictor scale. Applying the link function gives us predictions on the response scale (y=g−1η=g−1X ~β). A simulation-based approach also takes into account the situation where our summary is a non-linear function of the linear predictor (e.g., the log link function) (Wood, 2017, Section 7.2.6).

For example, to depict uncertainty geographically, we can perform a number of simulations (B) for the area of interest and summarize predictions on a per-grid-cell basis using the following algorithm:

1. For b = 1, …, B:

(a) Simulate from Nβ ˆ,Vβ ˆ, to obtain βb.

(b) Calculate predicted abundance for each prediction grid cell for this βb, N ˆb∗=g−1X ~βb.

(c) Store N ˆb∗, a vector of abundances, with one element per cell.

2. For each grid cell, calculate the empirical variance, percentiles, or statistic of interest for the N ˆb∗s.

In practice B does not have to be particularly large. Marra, Miller & Zanin (2012) achieved reasonable results with B = 100, though this is dependent on the summaries required. We note that here we talk about simply sampling from the multivariate normal distribution but in practice the approximation may not hold. In this case either importance sampling or a random-walk Metropolis–Hastings sampler can be used (the latter is implemented as gam.mh in the mgcv package).

Two further sources of model uncertainty from the GAM can be included in Vβ ˆ during model fitting and leave the above procedure unchanged. (i) Smoothing parameter uncertainty: estimated uncertainty in how wiggly terms should be, using the approach outlined in Wood, Pya & Säfken (2016), we then denote the covariance matrix as Vβ ˆλ ˆ. (ii) Model structural uncertainty: using shrinkage smoothers (either thin plate regression splines via the ts basis or cubic regression splines via the cs basis, in mgcv) that shrink model terms to zero (or near zero) effect size if necessary (Marra & Wood, 2011) as opposed to term selection via p-values or AIC. As terms are retained (but their effect sizes are shrunk), resulting uncertainty includes uncertainty about the model structure (conditional on the terms that were included).

Variability in environmental covariates

In dynamic environments that exhibit high environmental variability, a major contributor to uncertainty in estimates of abundance is the associated variability in population density due to movement of animals within, into, or out of the study area (e.g., Forney, 2000; Becker et al., 2014; Becker et al., 2019). To account for the potentially large changes that can occur within short periods, model predictions need to be made over temporally-relevant time periods, which may change depending on ecosystem dynamics (Mannocci et al., 2017a). For example, in a highly dynamic ecosystem such as the California Current (Bograd et al., 2009) environmental covariates (and hence model predictions) vary dramatically over time scales as short as several days during coastal upwelling events. Making predictions over multiple time periods allows the model to include different oceanic states and account for the variability in environmental covariates. One can think of this as being analogous to the procedure above but generating predictions from the realized (observed) distribution of possible environmental covariates, rather than the model parameters.

In ecosystems with limited seasonal or inter-annual variability or for models over larger areas and longer time periods, this environmental variability may be small, as illustrated for the Central North Pacific (Forney et al., 2015). Thus, both ecosystem factors and modelling scales will play a role in determining the relative importance of variability in environmental covariates.

Detectability

If we assume that detectability only varies at the segment level (e.g., with covariates representing weather conditions, e.g., Beaufort wind force scale), and not at the level of an individual observation, then we can use the variance propagation method of Bravington, Miller & Hedley (2021) to include the uncertainty about the detection function in Vβ ˆ (the GAM covariance matrix). Briefly, the method consists of refitting the GAM including a random effect that has the covariance from the detection function model (Vθ ˆ). We then obtain a modified covariance matrix, incorporating uncertainty in both detection function and GAM (Vβ ˆθ ˆ or Vβ ˆλ ˆθ ˆ if smoothing parameter uncertainty is also included). If detectability varies at the level of the observations (e.g., with group size or behavioural state), we may be able to apply the factor-smooth model of Bravington, Miller & Hedley (2021), categorising the detection covariate and then applying the variance propagation procedure. Finally, if detectability is constant (i.e., p ˆj=p ˆ∀j), we can simply apply the delta method. We do not recommend using the delta method in any other case, as then covariance between detectability and spatial model are ignored.

Trackline detectability

Relaxing the assumption that all objects on the trackline are observed (“g(0) = 1”) can be handled in a number of ways (Hiby, 1999; Barlow, 1995; Burt et al., 2014; Barlow, 2015). Most rely on having an additional observer or observers who are independent (or conditionally independent) of the main observer team. The proportion of observations missed by the main team can then be used to correct g(0). For the mark-recapture distance sampling method where g(0) is another component of the detection function model, the variance propagation approach can be used to include this uncertainty. Uncertainty around independent estimates of g(0) can be included via the delta method. Other methods for trackline detectability would need to be adapted appropriately.

Availability

If estimates of availability are constant in space/time (either calculated from observational studies or tags) then corresponding uncertainty can be included via the delta method (Ver Hoef et al., 2014). More complex models such as Borchers et al. (2013) and Borchers & Langrock (2015), which account for g0 ^ and availability simultaneously using auxiliary data from tags or a modified survey protocol, could be included in the GAM with some modification of the procedures in Bravington, Miller & Hedley (2021).

Other sources of uncertainty

Some sources of uncertainty do not have methods that can currently be applied in a spatial context, or have not yet been applied in the DSM literature. We discuss these briefly for completeness but do not address them fully here.

To account for measurement error in the environmental covariates, the procedure proposed by Stoklosa et al. (2015) to deal with both classical and Berkson-type measurement errors could be used. These require adaptation in how the spatial model (GAM in our case) is fitted.

When using a DSM, we have substituted the usual group size and encounter rate components that occur in distance sampling for the spatial model and as such these are handled as part of the GAM above. Group size uncertainty that arises from measurement error in observers’ counts needs to be addressed at the spatial model stage as it is likely that corrections may vary according to group size and sighting conditions, which vary spatially. Such errors will also likely effect estimates of response distribution hyper-parameters (such as overdispersion). Constant correction factors for group size uncertainty could be estimated depending on species and environment (Hodgson, Peel & Kelly, 2017), but this is currently an active area of research.

There are several methods to investigate species identification uncertainty. Methods that pro-rate species identity from additional data (e.g., Johnston et al., 2015) may not always be an option (though if they are, uncertainty can be included). Joint modelling or approaches where true species identity is a latent variable (Conn et al., 2013) might be one way to allow for this in the spatial model.

Combining uncertainty from multiple model components

We now focus on combining the above methods to create a single procedure for uncertainty estimation from complex DSMs (e.g., Eq. (2)). This modular workflow allows us to exclude any of the steps in the case where the model does not contain that uncertainty component. Our procedure is as follows (shown diagrammatically in Fig. 1):

Figure 1 Flow diagram showing our process for capturing uncertainty from multiple model components.

In this case we have a GAM model of spatial counts, a detection function and environmental variability to include in the model. Once detection function uncertainty is included in the GAM, some number (nb) posterior samples of model parameters (β ˆb) can be generated using the GAM-plus-other-components mean (β ˆ) and covariance matrix (Vβ ˆ,λ ˆ,θ ˆ here). These samples are used to generate potential density surfaces exploring both model parameter and environmental space. Finally, summaries can be calculated from these predictions.

1. Fit the DSM, with detectability, trackline detectability and/or availability included in the offset.

2. If any offset corrections are coincident, incorporate their uncertainty via the variance propagation method of Bravington, Miller & Hedley (2021).

3. Extract posterior estimates of the GAM parameters (β ˆ) and related covariance matrix (Vβ ˆ, Vβ ˆ,λ ˆ, Vβ ˆθ ˆ or Vβ ˆλ ˆθ ˆ; see ‘GAM (Smooths, Smoothing Parameters, Model Structure)’ and ‘Detectability’ for definitions).

4. Simulate B samples from βb∼Nβ ˆ,V∗ where V∗ is one of Vβ ˆ, Vβ ˆ,λ ˆ, Vβ ˆθ ˆ or Vβ ˆλ ˆθ ˆ.

5. For each time period that needs to be predicted (t = 1, …, T):

(a) Form the prediction matrix X ~t for this time period’s prediction grid.

(b) For b = 1, …, B posterior samples generated above:

i. Calculate predicted abundance for each prediction grid cell: N ˆb,t∗=g−1X ~tβb.

ii. Store N ˆb,t∗ for this iteration–time-period combination.

6. Summarize the per iteration–time-period predictions (N ˆb,t∗) by computing the appropriate summary statistic (typically mean or median) and the empirical standard error of the estimates.

7. Include uncertainty from non-coincident estimates via the delta method.

At point 5 above, we may wish to make spatial summaries: per-time-period abundances at each prediction location or average abundance across all time periods for each prediction location. We could also compute a time series of abundance. We show examples of these in the next section. We also note that the exact form of the variance estimators depends on the summary taken. Appendix S1 gives details of some common situations and their estimators.

The above algorithm can be implemented in any statistical programming language, and we include example code for the analysis in the next section with this article R. For step one we rely on functions in the R packages Distance and mrds for detection function fitting and dsm for fitting the DSM (examples of how to use these packages for model fitting are provided at http://examples.distancesampling.org). Step two can be accomplished using the function dsm_varprop in dsm. Step three relies on functions from the mgcv package, coef and vcov can be used to extract the parameter estimates and covariance matrix, respectively. To sample from the posterior distribution of the parameters we can use the rmvn function from mgcv. The remaining steps can be coded in base R and we recommend interested readers consult the code provided for examples.

Example: fin whales in the California current

To illustrate our approach, we use the example of fin whales in the California Current Ecosystem (CCE). Data were collected from line transect surveys conducted by NOAA’s Southwest Fisheries Science Center (SWFSC), July through early December of 1996, 2001, 2005, 2008, and 2014. Each of these surveys covered a broad area off the entire United States West Coast and, when combined, provided dense coverage of waters from the coast to approximately 556 km offshore (Fig. 2, left panel). Standardized line transect protocols were followed during all years using a team of three experienced observers stationed on the flying bridge of the ship (Barlow & Forney, 2007; Kinzey, Olson & Gerrodette, 2000). Our aims for this analysis are to estimate uncertainty around: (i) estimates of monthly abundance, (ii) a density map of fin whales in the study area, averaged over the whole time period, (iii) density maps of fin whales in the study area averaged within each year.

Figure 2 Raw data and results from applying our procedure to the fin whale data.

Top row, left: locations of segment centroids (black dots) for the fin whale survey with observation locations (orange dots) and underlying bathymetry. Top row, middle: predicted density for fin whales in the California Current Ecosystem. Top row, right: estimated standard error for the predictions, using the procedure we outline. Black dots give locations of observations of fin whales in the centre and right plots. Bottom row: confidence surfaces derived from a log-normal approximation; left is lower 2.5%, right is upper 97.5%.

The analysis presented here is designed to be as close to SWFSC’s modelling process as possible, to give an idea of how our framework can be adapted in practical situations. We refer readers to the technical memos and papers below for detailed information on this process. Our model consisted of the following components.

1. Detection function from Barlow, Ballance & Forney (2011). A half-normal detection function was fitted using the R package Distance (Miller et al., 2019) to all observations of large whales (Bryde’s, sei, blue, fin, humpback, and unidentified large whales), with covariates for ship (6 level factor), species (10 level factor), visibility, log average total school size for fin whales over all sightings and Beaufort (all continuous). Distances were truncated at 5.5 km. Fig. S1 shows the detection function.

2. Fixed estimate of g(0) from (Barlow & Forney, 2007, Table 3): g0 ^=0.921, CV =0.023. This estimated was based on the method developed by Barlow (1995) using conditionally independent observer data.

3. Dynamic environmental covariates taken from a 10 km resolution data-assimilative implementation of the Regional Ocean Modeling System (ROMS) in the CCE, which was produced by the University of California Santa Cruz Ocean Modeling and Data Assimilation group (Moore et al., 2011). The covariates sea surface temperature, sea surface height, mixed layer depth, and the standard deviation of each covariate within a 3 × 3 cell box surrounding each point were extracted at three-day intervals for the 5 years of sampled data. Bathymetric data (depth and standard deviation of depth) were derived from the ETOPO1 1-arc-min global relief model (Amante & Eakins, 2009).

4. A generalized additive model to describe the spatio-temporal variation in fin whale density as a function of a subset of the above environmental covariates. We used the same model structure and covariate selection procedure as Becker et al. (2016), yielding the final model: (3) Enj=Ajp ˆjg0 ^j expβ0+fxyxj,yj+fSSTSSTj+fSSTSDSSTSDj+fSSHSSHj+fMLDMLDj+fyearyearj,

where SSTSD is the standard deviation of sea surface temperature, SSH is sea surface height, MLD is mixed layer depth and yearj is the year in which segment j was surveyed. All smooths used a thin plate spline basis with shrinkage (Marra & Wood, 2011). fxy was constructed using a tensor product of smooths of longitude and latitude. Model term plots are shown in Fig. S3.

Predictions were made over grids of 11,860 cells covering the CCE, with a grid size of 0.09 degrees (≈ 10 km square). There were 821 grids giving daily values of covariates, 26 June through 6 December, for each of the 5 years, although occasionally the daily grids had missing grid cells because the covariates were not available from the ROMS model outputs. Based on the algorithm given in the previous section, we took the following steps to estimate uncertainty for Eq. (3).

1. Propagate detectability uncertainty into Eq. (3) via Bravington, Miller & Hedley (2021).

2. Extract β ˆ and Vβ ˆλ ˆθ ˆ.

3. Simulate 1000 samples from from Nβ ˆ,Vβ ˆλ ˆθ ˆ, to obtain βb for b = 1, …, 1, 000.

4. For each time period that needs to be predicted (t = 1, …, 821):

(a) Form the prediction matrix X ~t for this time period, t.

(b) For b = 1, …, 1, 000 posterior samples:

i. Predict abundance for each prediction grid cell: N ˆb,t∗=g−1X ~tβb.

ii. Store N ˆb,t∗.

5. Calculate per-prediction-cell means and variances over N ˆb,t∗ for all years and for each month.

6. Calculate monthly means and for N ˆb,t∗ to obtain abundance time series.

7. Include g0 ^ uncertainty via the delta method.

This procedure encapsulates spatial model uncertainty, detection function uncertainty, trackline detection uncertainty and spatial covariate variability.

Resulting maps of overall predictions and corresponding standard errors are shown in Fig. 2. Yearly maps (Fig. S2) and monthly time series plots of abundance (Fig. 3) showed increasing abundance between years (in agreement with e.g., Moore & Barlow, 2011). Uncertainty appeared to be highest in 2008 and on June 26 in particular; further investigation showed that this was due to large mixed layer depth values (Fig. S4). On June 26, 2008 the mixed layer depth reached 137 m, whereas in the data used to build the model the maximum was about 100 m, at which point the smooth showed increasing behaviour (Fig. S3, bottom row left panel). There are a number of options for modellers when this kind of extrapolation occurs, including excluding the cell from predictions, winsorizing or leaving as-is; Bouchet et al. (2020) provide an overview and practical tools for DSMs.

Figure 3 Time series of monthly abundance estimates for fin whales in the California Current Ecosystem.

Dots indicate mean abundance estimates over the days/iterations and lines 95% intervals based on a log-normal approximation.

The data used to fit models here had some overlap with those used in Becker et al. (2016), but for a more direct comparison Figs. S5 and S6 show plots comparable to Fig. 2 and Fig. S2 but where only environmental uncertainty was included in the variance calculation procedure (as was the case in Becker et al., 2016). We note that when only environmental variability is accounted for, plots seem to show much less detail in spatial uncertainty patterns.

Similar analyses of these data were provided by Moore & Barlow (2011) and Nadeem et al. (2016). Both provided a state-space model for abundance and the latter used a spatial (only over four large areas) Ricker model to obtain abundance estimates. Neither provided spatially-explicit estimates; only stratified abundance was calculated. Detection probability and trackline detection (using estimates from Barlow & Forney, 2007), along with their associated uncertainties were included in estimates of abundance. For comparison (Fig. 4), we calculated summary abundance estimates at the yearly level, along with 95% intervals (based on a log-normal approximation).

Figure 4 Comparison of published yearly estimates of fin whale abundance in the California Current Ecosystem.

Dots show abundance estimates and lines their 95% intervals.

Discussion

Maps of point estimates (e.g., mean density over a time period) are often given prominence in articles and reports but without some measure of uncertainty they are not useful for conservation (Jansen et al., 2022). In many cases, only uncertainty from the spatial model is reported in studies, perhaps because it could be obtained as a convenient output directly from that modelling stage, and modellers lacked ready methods for considering other sources of uncertainty. The framework we provide here allows modellers to estimate and evaluate these sources of uncertainty in their DSMs. These approaches can be adapted to other spatial modelling methodologies.

We see the delta method as the last resort when we are not able to include that uncertainty via other methods. We do not however want to discourage people from using this method if that is all that is available to them. A significant advantage of incorporating covariances via Bravington, Miller & Hedley (2021) where possible over using the delta method is that uncertainty estimates can go down as well as up (e.g., when there is negative covariance between the detection function and spatial model).

A practical benefit to using the simulation-based approach we outline here is the computation is easily parallelizable; speeding it up is simply a matter of splitting the simulation runs across a larger number of cores. Storage and retrieval of interim results may prove to be more of an issue; in our example we switched to using serialized objects (via R’s saveRDS functionality) over text-based (comma separated value) files to improve the speed of final calculations and reduce disk space, though storing in this format still used about 68GB of disk space. More efficient binary storage options could be investigated for larger prediction grids. Iteration over time period is conceptual and was done for a practical computing consideration; we could have created a prediction matrix of all time periods at once ( X ~ rather than X ~t), but this may not fit in memory so we formed one matrix for each time period. We applied the online variance calculation method of Welford (1962) post hoc. This approach could be included in the algorithms above, avoiding storage of N ˆb,t∗. Appendix S1 also provides equivalent analytic expressions for the variances in some useful situations. We find the simulation framework easier to understand and reason about, so that is the main focus here.

A popular reply to the question of how to calculate uncertainty for complex models is “do a bootstrap”. What is usually meant by this is that one should resample the data with replacement, refitting models each time and calculating some summary statistic (a non-parametric bootstrap). This seems appealing but it has been shown that the use of so-called “naïve” bootstraps leads to underestimation of uncertainty. The issue stems from the fact that GAM terms are structured random effects, which have priors on them. When data are resampled, the structure (prior) is ignored so uncertainty is collapsed leaving only the sampling variation in the bootstrap resamples. Bravington, Miller & Hedley (2021) show a simple simulated example of this happening. A secondary issue is that of coming up with a resampling scheme for spatial uneven data. A sufficiently complicated bootstrap might be constructed to achieve our aims here, but we have instead focussed on an easily-implementable, modular approach.

We have proposed a framework for uncertainty estimation to allow for the inclusion of relevant sources of uncertainty in final estimates. We do not believe that all of these sources should be included in all models, at least in part because the corresponding mean model components may not be necessary (e.g., availability may not be an issue for large terrestrial mammals). There are situations in which we think the environmental variability may not be necessary (models with non-dynamic covariates) or may be less of an issue (less dynamic system), so this component can be easily omitted (or included and tested). We also note that there are multiple approaches possible for each source of uncertainty and for many there is no best practice at the moment, so our intention is not to be prescriptive. Our hope is that this framework will prompt more discussion of the issue of how to estimate these quantities, since we can now include their uncertainty final estimates.

Supplemental Information

Supplemental Information 1 Histograms of observed distances with detection functions overplotted

Black lines show average detection functions. Coloured lines give the detection function varying the given covariate with other covariates fixed. These fixed were vessel (segment mode) “McArthur II”, visibility (segment median) 5.67, school size 1, Beaufort (segment median) and species set to fin whale.

Click here for additional data file.

Supplemental Information 2 Yearly predicted densities and standard errors

Top row: predicted yearly densities for fin whales in the California Current Ecosystem. Bottom row: yearly estimated standard errors for the predictions, using our procedure. Both measures are plotted on the same scale for easy comparison. Black dots give locations of observations of fin whales.

Click here for additional data file.

Supplemental Information 3 Plots of fitted smooth terms for the fin whale model

Top row and bottom row left and centre show univariate effects of the covariates, grey bands give the mean estimates +/ − 2 standard errors. The effective degrees of freedom are given in brackets on the vertical axes and rug plots show data locations. Bottom right plot shows the two dimensional tensor product spatial smooth with 15.41 effective degrees of freedom.

Click here for additional data file.

Supplemental Information 4 Predictions on the linear predictor scale of predictions per covariate for 2008-6-26

This was when there were large observed values of mixed layer depth, causing the large uncertainty in Figure 3. Plots are directly comparable in terms of their influence and were generated using the function plot_pred_by_term in the R package dsm.

Click here for additional data file.

Supplemental Information 5 Comparison when only environmental uncertainty was included in our procedure

Left: Estimated standard error for the predictions, using only environmental uncertainty. Dots give locations of observations of fin whales. Right: differences between the full uncertainty in Figure 2 (right panel) and the left plot here. Note that as expected the uncertainty is always larger for the full uncertainty procedure.

Click here for additional data file.

Supplemental Information 6 Yearly comparison when only environmental uncertainty was included in our procedure

Top row: yearly estimated standard errors for the predictions, using only environmental uncertainty. Dots give locations of observations of fin whales. Bottom row: differences between the full uncertainty in Figure S2 and the top row here. Note that as expected the uncertainty is always larger for the full uncertainty procedure.

Click here for additional data file.

Appendix S1 Making predictions and estimating variance at and over multiple time points

Click here for additional data file.

The survey data used in this analysis were collected by a dedicated team from NOAA’s Southwest Fisheries Science Center and we thank everyone who contributed to collecting these data. We also thank the UCSC Ocean Modelling Group for providing ROMS output.

Additional Information and Declarations

Competing Interests

Author Contributions

Data Availability

Elizabeth A. Becker is employed by Ocean Associates Inc.

David L. Miller conceived and designed the experiments, performed the experiments, analyzed the data, prepared figures and/or tables, authored or reviewed drafts of the article, and approved the final draft.

Elizabeth A. Becker conceived and designed the experiments, performed the experiments, analyzed the data, authored or reviewed drafts of the article, and approved the final draft.

Karin A. Forney conceived and designed the experiments, authored or reviewed drafts of the article, and approved the final draft.

Jason J. Roberts conceived and designed the experiments, performed the experiments, authored or reviewed drafts of the article, and approved the final draft.

Ana Cañadas conceived and designed the experiments, authored or reviewed drafts of the article, and approved the final draft.

Robert S. Schick conceived and designed the experiments, authored or reviewed drafts of the article, and approved the final draft.

The following information was supplied regarding data availability:

The code and small data files are available at Github: https://github.com/densitymodelling/lemur_fin_analysis/.

The processed ROMS grids are available at figshare: Miller, David; Becker, Elizabeth (2022): ROMS grids for predictions. figshare. Dataset. https://doi.org/10.6084/m9.figshare.20132291.v1.

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
