# Peer review of "Estimating uncertainty in density surface models"

_PeerJ, doi:10.7717/peerj.13950_

## Round 0.1 · original submission · Minor Revisions

This was a great paper to read; both reviewers thought the article is well written and very timely. They each include a handful of suggestions that I think will improve the paper. Reviewer 2 noted that having reproducible scripts (even with dummy data) might make this approach more widely used.

My one comment was whether preferential survey sampling and/or shifts in survey effort across seasons or space might be considered measurement error here, or additional sources of uncertainty?

·

Basic reporting

No comment

Experimental design

No comment

Validity of the findings

No comment

Additional comments

This is a very well written, streamlined and to-the-point presentation of an important proposed method for incorporating multiple sources of uncertainty into spatial GAMs / density surface models. I have only minor comments, and think it’s especially great that the authors provided reproducible code and examples on GitHub so that researchers can adapt this method to fit their data/systems/needs without having to start from scratch. Nice job.

L40-42: I think you can be a bit more specific about how including uncertainty allows managers to make good decisions (rather than simply stating it allows them to make the best possible decisions / avoid strategies that have dire consequences). E.g. ‘…allows managers and policy makers to evaluate the level of confidence associated with estimates of distribution and abundance and therefore the probability that a proposed management action will have the desired outcome.’ (Or however you want to lay it out, but be more specific).

L105: Suggest a period between others and we: ‘components than others. We begin..’ Also believe this is supposed to read ‘by looking at those’ (not ‘that those’).

L113: Can you provide a citation for this? I was not aware that mgcv / REML was equivalent to empirical Bayes estimates… likely there will be other readers who are also not aware of this.

L114: What if your B-hats aren’t normally distributed (as is often the case in true posterior distributions)? I believe mgcv just provides maximum likelihood and SE for model estimates, so I think you should at least state here that one assumption is that those are normally distributed ‘approximate posteriors’. Also suggest you define V as the covariance matrix here (you do so later in the text but it was a lingering question until that point).

Figure 2: Left panel seems a little odd to have shallow water in darkest blue and deep water in lightest yellow. Might make it easier to see the orange sighting locations if the depth scale were reversed. Also, one of the stated goals of this method was to make it easier for policymakers to interpret uncertainty and make decisions based on that. In the right panel showing prediction SEs, you can see where uncertainty is high but not what the result of that uncertainty on possible whale densities is. With decisionmakers in mind, it might also be nice to have two plots showing the 95% (or some other interval) highest and lowest predicted densities from the uncertainty simulations for each grid cell. That would demonstrate where the possible areas of highest (and lowest) density are, which would be a nice additional companion to the predicted and SE plots.

L302: SDM as in species distribution modeling, or a typo for DSM?

L312: speed ‘of’ final calculations?

·

Basic reporting

The manuscript provides much-needed guidance on how to characterize and combine sources of uncertainty in density surface models (DSMs), in the form of a broad modular framework that can be adapted to different situations. I do not have the statistical expertise to comment on the mathematical aspects of the manuscript, but as a frequent end-user of the scripts and packages used for distance sampling and DSM estimation, I focus my review on the clarity and usefulness of the paper’s contents for DSM practitioners.

The language of the manuscript is concise and professional throughout and its overall structure is especially clear, with a brief overview of DSMs and a list of the various sources of uncertainty followed by an overall summary of the approach which is then applied to a concrete example. However, given the complexity of the subject matter, I have made some suggestions as to how some of these sections’ structure or some of their specific language can be made even clearer.

A. Structure and content of the introduction: in the first paragraph of the introduction, the authors explain in very broad terms why estimating uncertainty allows managers to make better decisions. Then, the authors give more specific justification for their interest in this topic in the third paragraph (MMPA). I would suggest that the introduction would flow more logically if this third paragraph (lines 48-58) was placed directly after the first one, and that lines 43-47 were moved down and merged with lines 59-61 (since they both describe the approach).

Also, I know that there is no room here for a thorough summary of the rich literature on the topic of uncertainty, but would suggest that at least one example of a quantitative use of the uncertainty around abundance estimates be mentioned in the “MMPA” paragraph: a lot of approaches based on the precautionary principle make explicit use of the variance of an estimate, such as PBR which is calculated not based on a point/mean estimate but rather on a 20% quantile of the estimate (therefore, changing the uncertainty/variance around an estimate directly changes the number of “takes”).

I would drop the use of the word “systematic” in line 43, which is both ambiguous in the context of survey designs and also slightly at odds with the use of the more appropriate word “modular” to describe the approach.

Finally, I would add a few lines to better define “simulation-based”, as some readers might be more familiar with calling or perceiving this as a form of “resampling”. The first paragraph of p.3 of Appendix A (under “Simulation”) has very useful language that could be easily integrated here without making the main text too complex, but would better prepare the reader for the rest of the framework’s overall philosophy.

B. Section 3 provides a very useful list of all the model components that can generate uncertainty. As such, I feel that renaming this section’s heading to include explicitly that this is a list rather than the general approach (which is given in section 4) would make it clearer at first glance. For instance, something like “Characterising uncertainty for model components” or “Model components and uncertainty estimation”.

I also wonder if it would be more logical to reorder the components/sources of uncertainty in the order of most analyses (including the fin whale analysis given here as an example and in the flow chart), i.e., starting with Detectability and Trackline detectability, Availability, and then moving on to GAMs and environmental covariates. This would better fit the analytical flow of sections 4 and 5.

Another comment for this section is that Group size is a common source of uncertainty for some species (to the point that the Distance software reports a breakdown of variance for design-based analyses into 3 components: detection function, encounter rates and group size). Here, group size is briefly mentioned in the other sources of uncertainty but left me confused with where and how it should be treated. Line 189 states that “Group size uncertainty needs to be addressed at the spatial model level” but in the fin whale example, group size is a covariate in the detection function (as is often the case). Does that mean that group size will be present in both detection function and spatial model components (i.e., coincident) for a lot of analyses and if so, how should it be dealt with?

I feel that group size might deserve its own numbered paragraph in this section and some clearer guidance on how to include it, even if it’s not an important part of the fin whale example (e.g., depending on whether there are reasons to believe that group size varies spatially).

C. Section 4 provides a brief overview of the heart of the paper, i.e. the overall framework to combine uncertainty from multiple model components. I appreciate its brevity and how light it is in terms of mathematical formulas – it reads more like an overall recipe than a detailed walkthrough, and that’s a good strategy given that there is more information in the Appendix A (which is a very useful description of various estimators) and in the fin whale example. However, I was then a bit surprised at the lack of any more practical guidance on how to achieve each of the specific steps in the workflow. I feel that, if I wanted to use this workflow for my own data and analyses, I would benefit from a bit of more specific guidance in terms of how to perform the least obvious steps (especially 3 and 4), for instance things like which parts can be achieved with existing R functions in the Distance, dsm, mcgv packages, etc. I’m not sure how to include this additional information without disrupting the simplicity of the overall workflow presentation – perhaps as a supplemental appendix, or maybe by adding a couple of sentences to the paragraph 216-220?

Experimental design

No comment.

Validity of the findings

I note that R code has been made available via the first author’s github page (something that could be better emphasized in the text, but maybe this will be made more obvious in the final journal layout). The scripts are hard-coded, i.e. specific to this particular analysis (as opposed to a set of functions/packages that can apply to any analysis), which will require solid understanding on the part of anybody who wants to adapt it to their needs, but having access to those is still extraordinarily useful. However, the raw data do not seem to be available (i.e., no data/Bp_ModelingSegs_BF0_5_CCE_1996_2014_no09_5km.csv and ddf Object LgBaleenWh_1991-2018_BEAUF 0-6.RData objects in the data file as far as I can see), which will limit somewhat the ability of users to reproduce and adapt the method to their own needs/datasets.

---

## Round 0.2 · accepted · Accept

Thanks for addressing all of the comments from the reviewers - we appreciate your work on this paper.